# Evaluation of Oral Mucosal Lesions Using the IllumiScan^®^ Fluorescence Visualisation Device: Distinguishing Squamous Cell Carcinoma

**DOI:** 10.3390/ijerph191610414

**Published:** 2022-08-21

**Authors:** Yuki Taguchi, Shigeaki Toratani, Kensaku Matsui, Seiya Hayashi, Natsuki Eboshida, Atsuko Hamada, Nanako Ito, Fumitaka Obayashi, Naohiro Kimura, Souichi Yanamoto

**Affiliations:** 1Department of Oral Oncology, Graduate School of Biomedical and Health Sciences, Hiroshima University, 1-2-3 Kasumi Minami-ku, Hiroshima 734-8551, Japan; 2Department of Dentistry and Oral Surgery, Hiroshima Prefectural Hospital, Hiroshima 734-8530, Japan; 3Department of Dentistry and Oral Surgery, JA Onomichi General Hospital, Onomichi 722-8508, Japan

**Keywords:** oral squamous cell carcinoma, autofluorescence, fluorescence visualisation device, oral mucosal lesion, epithelial dysplasia, coefficient of variation

## Abstract

We evaluated whether fluorescence intensity (FI) and its coefficient of variation (CV) can be used to diagnose squamous cell carcinoma (SCC) through IllumiScan^®^, an oral mucosa fluorescence visualisation (FV) device. Overall, 190 patients with oral mucosal lesions (OMLs; SCC, 59; non-SCC OMLs, 131) and 49 patients with normal oral mucosa (NOM) were enrolled between January 2019 and March 2021. The FI of the images was analysed using image analysis software. After establishing regions of interest for SCC, non-SCC, and NOM, the average FI, standard deviation (SD), and CV were compared. There was a significant difference in the average FI for all pairs of comparisons. The SD was not significantly different between the SCC and NOM groups (*p* = 0.07). The CV differed significantly for NOM (*p* < 0.001) and non-SCC groups (*p* < 0.001) relative to the SCC group but was not different between NOM and non-SCC groups (*p* = 0.15). Univariate analysis of SCC and non-SCC groups showed significant differences for all factors, except age. However, multivariate analysis showed a significant intergroup difference only in the CV (*p* = 0.038). Therefore, analysing the CV in FV images of OML may be useful for the diagnosis of oral cancer.

## 1. Introduction

Oral cancer is relatively easy to diagnose because it is more easily visualised and palpated than other organs. However, differentiating oral cancer from a variety of mucosal lesions, such as leukoplakia, erythroplakia, lichen planus, and refractory stomatitis, is difficult. Diagnoses of oral squamous cell carcinoma (SCC) and oral potentially malignant disorder (OPMD) are based on a combination of biological staining with iodine [1] and toluidine blue [2], scraped cytology [3], fluorescence visualisation (FV) [4], and/or tissue biopsy. On iodine staining, a brown colour represents lesion negativity, while the absence of stain retention represents positivity [1]. The contrast between normal and dysplastic lesioned areas can be easily distinguished visually. However, these approaches have drawbacks, such as restrictions on the use of iodine in iodine-allergic patients and lack of staining of keratinised epithelium, e.g., the gingiva [5]. Scraped cytology is simple, relatively non-invasive, and allows epithelial cell analysis after scraping the oral mucosal surface [6].

However, this method is unreliable because it does not access deep cells [7]. It has been pointed out that biopsy, considered the best method for diagnosing oral cancer, is invasive and sometimes results in tumour cells being seeded into the surrounding tissues [8]. 

Since the 1980s, FV devices have been used to detect oral cancer [9]. These devices utilise the excitation of endogenous fluorophores, such as certain amino acids, metabolic products, and structural proteins, by an extrinsic light source. Within the oral mucosa, the most relevant fluorophores are epithelial nicotinamide adenine dinucleotide (NADH), flavin adenine dinucleotide (FAD), and stromal cross-linked collagen [4]. The fluorophores absorb photons from the exogenous light source and emit lower-energy photons, presenting as fluorescence [10].

In cancer tissues, energy metabolism switches from oxidative phosphorylation to aerobic energy production [11], decreasing NADH and FAD. Therefore, autofluorescence is attenuated in cancer tissues compared to normal mucosa [12]. Autofluorescence is also attenuated and loss of FV (FVL) is observed in cancerous tissues due to the disruption of collagen structural proteins and elastin structures in the normal mucosal stroma. Furthermore, in areas of angiogenesis and inflammation within a cancer tissue, autofluorescence is also attenuated due to significant absorption of light by haemoglobin [13]. FVL can be observed by (i) a decrease in NADH and FAD, (ii) destruction of collagen structural proteins and elastin structures, and (iii) absorption of light by haemoglobin in areas of angiogenesis and inflammation. By observing the autofluorescence of the normal mucosa and the attenuated autofluorescence in cancerous lesions, the status and extent of lesions can be evaluated [14,15,16].

Although many devices have been developed and commercialised to evaluate tissue autofluorescence, the VELscope^®^ (LED Dental Inc., Vancouver, BC, Canada) was the first commercial auto-focus imaging device approved for oral use in 2008 [4,17]. The IllumiScan^®^ (Shoufu, Kyoto, Japan) (Appendix A) is an FV device that emits a 425 nm blue light and receives images of the fluorescence generated by the mucosa [18,19]. The light received includes autofluorescence from the mucosa, reflected light, and red fluorescence from bacteria. Light other than green light (490–550 nm) is eliminated before visualising the image on a monitor (Appendix A). This device is almost as easy to use as is a digital camera and is highly sensitive to oral mucosal lesions (OMLs) that could be missed easily by visual examination (Appendix A). Furthermore, it is suitable for the long-term follow-up of mucosal lesions because it is a non-contact, non-invasive, quick, and easy method that can be used repeatedly.

However, the auxiliary tools for OML detection, such as IllumiScan^®^ and VELscope^®^, have been reported to have high sensitivity but low specificity [20]. This is because irradiation of cancerous and inflammatory tissue with blue excitation light results in attenuated autofluorescence in both, making it difficult to visually distinguish between the two using FVL (Appendix A).

Furthermore, keratinised areas have enhanced fluorescence intensity (FI). Keratinisation is often present on cancer surfaces, including that in SCC, and in epithelial dysplasia. This causes an improved FI and obscures the FVL of these lesions (Appendix A), making it difficult to find a significant difference in the average FI of IllumiScan^®^ images between SCC and non-SCC tissues.

The advantage of autofluorescence devices is that they enable the direct visualisation of tissue fluorescence; thus, there are many reports of visual recording and subjective judgment [21,22]. For example, Ganga et al. visually classified lesions into two groups: a group containing lesions that showed FVL, appeared dark with pale green autofluorescence compared to surrounding unchanged tissue, and showed malignant or dysplastic changes and another group that showed fluorescence visualisation retention (FVR) and no autofluorescent changes compared to surrounding unchanged tissue [20]. Meleti et al. subjectively classified lesions as normofluorescent, hypofluorescent, or hyperfluorescent [18]. Wang et al., after determining FVL, FVR, and fluorescence visualisation increase (FVI), further classified them according to autofluorescence pattern into FVR, FVL + FVR, FVI, and FVI + FVL [17].

In this study, we attempted to measure the luminance value of the fluorescence of OML, quantify it, and analyse it. The luminance value of healthy mucosa in the same area as the lesion was measured and compared with that of the lesion. Furthermore, we evaluated whether FI and its variation, i.e., the coefficient of variation (CV), could be used when analysing IllumiScan^®^ images to diagnose SCC.

## 2. Materials and Methods

Of the patients who visited the Department of Oral Oncology at Hiroshima University Hospital from January 2019 to March 2021, 239 patients were included in the study. Of the 239 cases, 190 cases of OMLs were identified during conventional oral examination under white light and were found to have features compatible with the clinical suspicion of OPMD or early-stage SCC. Inclusion criteria were patients who gave informed consent for imaging, were imaged prior to biopsy, and had a definitive diagnosis confirmed by biopsy. Patients with lesions of T3 or higher as per the International Union Against Cancer (UICC) 8th edition TMN malignancy classification and patients without adequate images due to aperture defects were excluded from this study to reduce the burden on patients [23]. The 49 cases of the normal oral mucosa (NOM) were patients who visited the department for temporomandibular joint disorders or other disorders and volunteered to be photographed, and areas of the oral mucosa without clinical abnormalities were photographed.

The lesions were recorded using a digital camera (Nikon, Tokyo D5500, Tokyo, Japan) under white light and were then photographed using IllumiScan^®^ in a dark room. The brightness of the irradiation light was maintained and did not vary between cases. The distance between the IllumiScan^®^ lens and lesion was approximately 30 mm.

IllumiScan^®^ images were transferred to a computer hard drive and analysed using ImageJ version 1.53 (NIH, Bethesda, MD, USA). Each image was divided into RGB components (R: red, G: green, B: blue), and a black-and-white image (G-image, Appendix A) was created from the green component image. A region of interest (ROI, solid yellow line in Appendix A) was defined in the lesion area of the G-image based on intraoral photographs and intraoral findings by the attending physician. The ROIs of the NOM areas were set at the sublingual mucosa, buccal mucosa, the transition area from the molar attachment gingiva to the buccal mucosa and the mucosa on the floor of the oral cavity [24,25]. The NOM area was selected to be wide enough to ensure that the target lesion was at a right angle to the irradiation light. FI was expressed at 256 levels (0–255). The FI of the ROI was measured, and after calculating the average FI, the standard deviation (SD) and CV (SD/average FI) × 10 were calculated.

This study was approved by the Ethical Review Committee of Hiroshima University Hospital (approval No. C-129-2).

### Statistical Analysis

Statistical analysis was performed using JMP^®^ 16 software (SAS Institute Inc., Cary, NC, USA). The Dunn’s test was used for non-parametric comparisons between all pairs of NOM, SCC, and non-SCC groups, with regard to luminance, SD, and CV. Fisher’s exact probability test was used to compare sex differences. Multivariate analysis was performed using nominal logistic analysis. A *p*-value < 0.05 was considered statistically significant. The area under the curve (AUC), threshold, sensitivity, and specificity were calculated for average FI, SD, and CV, using receiver operating characteristic (ROC) curve analysis. The cut-off values were defined based on the Youden index.

## 3. Results

The clinicopathologic characteristics of the patients are shown in Table 1. Of the 239 cases, 59 were of SCC, 131 were of non-SCC lesions, and 49 were of NOM. The pathological diagnoses of non-SCC were lichen planus in 64 cases, epithelial dysplasia in 22, and carcinoma in situ in 16 cases. The male/female ratio was 35/24 for SCC, 50/81 for non-SCC, and 30/19 for NOM. The median age of the participants was 75 years among the SCC cases, 71 years among the non-SCC cases, and 36 years among the NOM cases. Topographically, 84.8% of SCC cases occurred in the gingiva (n = 29) and tongue (n = 21). In the non-SCC cases, 72.6% of lesions developed on the buccal mucosa (n = 53) and gingiva (n = 42). Of the NOM images, 53.1% (n = 26) were of the tongue.

The relationships between NOM, SCC, and non-SCC for average FI, SD, and CV are shown in Figure 1. The NOM average FI was statistically significantly higher than in the SCC and non-SCC groups (*p* < 0.001). The FI was significantly lower in the SCC group than in the non-SCC group (*p* = 0.013). The SD in the non-SCC group was significantly lower than in the NOM and SCC groups (*p* < 0.001), but the SD in the SCC and NOM groups were similar (*p* = 0.071). CV was significantly higher for the SCC group than for the other two groups (*p* < 0.001) but similar between the non-SCC and NOM groups (*p* = 0.15).

Univariate analysis of the SCC and non-SCC groups for sex, age, average FI, SD, and CV showed significant differences in all items, except age. On the other hand, multivariate analysis revealed that only the CV (*p* = 0.038, Odds ratio = 13, 95% CI: 1.09–154.7) was independently significant (Table 2).

The ROC curves for differentiating SCC from non-SCC and NOM are shown in Figure 2. The AUC of the average FI was 0.72. Assuming an average FI cut-off of 98.5, the sensitivity and specificity were 71.2% and 67.8%, respectively (*p* < 0.0001). The AUC of the SD was 0.63. Assuming an SD cut-off of 11.2, the sensitivity and specificity were 83.1% and 65.6%, respectively (*p* < 0.0001). The AUC of the CV was 0.84. Assuming a CV cut-off of 1.70, the sensitivity and specificity were 72.9% and 78.3%, respectively (*p* < 0.0001).

## 4. Discussion

We obtained FV images with IllumiScan^®^ to observe OMLs and investigated image evaluation methods for the detection of oral cancer. While the SD was not significantly different between the SCC and NOM groups, the CV of SCC differed significantly from that of NOM (*p* < 0.0001) and non-SCC (*p* < 0.0001) but not between NOM and non-SCC (*p* = 0.15). In multivariate analysis, only the CV (*p* = 0.038) was a significant factor in distinguishing SCC. Therefore, the analysis of the CV in FV images of OML obtained with an FV device may facilitate oral cancer diagnosis.

Tissue autofluorescence is used for diagnosis as a screening for precancerous lesions in the lungs, cervix, skin, etc. [21]. In this technique, the lesion is irradiated with light of a specific wavelength, fluorescence is emitted from the normal tissue, and the light received is imaged. Since cancer and other tissues absorb irradiated light, lesions located in normal tissues can be visually distinguished. The 5-year survival rate for oral cancer in Japan is 86.6% for localised cancers [26], and early detection can markedly improve prognosis. Therefore, a simple device that can be routinely used in clinical practice to observe the oral mucosa is needed. IllumiScan^®^ induces autofluorescence of the NOM and allows image-based detection of the attenuated fluorescence of cancerous tissue and epithelial dysplasia. The imaging is non-invasive and repeatable. It is considered easy to operate and does not require specialised training. However, the use of the device by unskilled users can lead to high false-positive results because of misinterpretation. Furthermore, it has been reported that the device should not be used at the screening stage as false positives may lead to unnecessary referrals and biopsies [27,28]. However, there have been reports of an increase in specificity from 8% to 10% when the FV device was used in addition to conventional oral cavity examination for the detection of epithelial dysplasia and OPMDs, even if the clinician was not specifically trained in the use of the FV device [29]. Some reports point to the usefulness of repeatable imaging in high-risk patients who require a longer follow-up period [30].

Therefore, this technique may be useful as a clinical diagnostic aid to detect SCC and other lesions that are often missed and could guide biopsy. Further, it could be used not only for cancer screening but also for long-term follow-up of precancerous lesions, preoperative determination of the resection area [31], and post-operative follow-up.

More than 15 years have passed since the introduction of FV devices, and they are now approved and marketed as medical devices. It has been reported that the oral mucosal observation system is an excellent tool for detecting OMLs [32,33] and can depict stomatitis that may be overlooked during visual examination. The average FI of SCC is significantly lower than that of NOM due to FVL in SCC. However, considering the individual average FI, some SCC and non-SCC lesions had a high FI, comparable to that of NOM. In a report examining histological characteristics, the epithelium of malignant lesions with a thick keratin layer was reported to be hyperfluorescent, while lesions with a thin keratin layer were dark and hypofluorescent [18]. The FI value of SCC was thought to capture the thickness of keratin, not just the attenuation of autofluorescence. Furthermore, considering that hyperplasia and increased keratinisation are frequently observed in OPMD such as leukoplakia and in highly differentiated SCC, the FI value could also indicate the differentiation level of oral squamous cell carcinoma (OSCC), since it has been reported that highly fluorescent lesions are in the early stages of malignant transformation, while hypofluorescent lesions are found at the undifferentiated cancer-like stage [34].

Molecules that can generate autofluorescence with a certain light include tryptophan, porphyrins, collagen cross-linkers, elastin, NADH, and FAD. Conversely, they also lose fluorescence and appear darker. Fluorescence loss may occur because of increased metabolic activity of the epithelium, disruption of collagen cross-links, increased tissue blood volume, and the presence of pigment. Haemoglobin, pigmentation, and metal tattoos are observed as FVL because they absorb light. Benign lymphoid tissue aggregates due to the prominent vascular melanin seen in mild trauma and inflammation and the absence of collagen and leukocytes can be observed as FVL due to the lack of autofluorescence [21]. In this study, some non-SCC lesions showed low FI, equivalent to that of SCC. To overcome this problem, some reports have attempted to reduce the number of false positives by performing a 2-week follow-up. However, reportedly, even after a 2-week follow-up, the specificity of the VELscope^®^ did not improve, so, it was suggested that the 2-week follow-up should not be performed for high-risk lesions [35]. Inflammation attenuates fluorescence. To rule out inflammation, careful history review is needed and the response to follow-up and anti-inflammatory treatment should be considered.

Because FI depends on various mucosal conditions, it has been reported that FV systems have high sensitivity but low specificity [8]. This is because FV devices are not capable of discriminating between malignant and benign lesions and inflammation [32]. One of the reasons for this is as follows. The CV of FI in images may indicate intra-tumour heterogeneity [19,36]. Kosugi et al. [37] quantitatively evaluated the changes in FI during progression in the transformation from normal epithelium to SCC in rats and reported that the FI variability increased with progression. Masuda et al. reported that fluctuations in FI can distinguish normal epithelium from low-grade dysplasia [38]. Kosakai et al. reported that SCC showed an uneven luminosity, while OLP showed a uniform decrease in luminosity [39]. The SD and CV indicate variability. The SD tends to be larger with a higher average FI, but the CV divides the SD by the average FI, so that this variation is not affected by the FI values. We believe that the SD of NOM was larger than that of SCC because the average FI of NOM was higher. The high CV of SCC and the lack of a significant difference in the CV between NOM and non-SCC suggest that the CV may facilitate an SCC diagnosis. This is also supported by the fact that the CV was independently significant in the multivariate analysis [40].

Other institutions have also reported the data obtained with IllumiScan^®^ (Appendix A) [19,41,42,43]. The difference in FI values among institutions was marked but was thought to be influenced by environmental factors (differences in ambient light and light output) related to light exposure. In addition, the FI measurement for control is set outside the lesion, so the measurement area is small. One of the shortcomings of IllumiScan^®^ is that compared to the centre of the irradiated field, the image is darker on the outside because the excitation light is not emitted perpendicularly. In this study, the NOM was at the centre of the irradiated field, and the area evaluated is in the same region as the lesion. Although the FI values appear to differ, the trend of FI values according to the lesion was similar, i.e., the FI value of SCC was low, followed by that of inflammatory lesions and OPMD, such as oral lichen planus and epithelial dysplasia, and NOM, respectively. Additionally, the CV for SCC was found to be approximately 0.2–0.3 at other institutions and was higher than those for OPMD and NOM. Therefore, the CV can be considered as a general-purpose value that eliminates differences caused by environmental factors [40].

In a 2010 recommendation, the American Dental Association (ADA) noted that the VELscope^®^ FV device was useful for determining surgical margins and biopsy sites [44]. Additionally, FV devices using probes, such as 5-Aminolevulinic acid, hypericin, aminopeptidase, and rhodamine, have been reported [32]. The combination of autofluorescence and fluorescence probes is inexpensive and can provide an accurate diagnosis of oral cancer to assist dentists during daily clinical activities [8]. However, there is also a reasonable time constraint to move the probe into the lesion. In the future, it is expected that imaging with the FV device will be used in conjunction with intraoral examinations for post-operative follow-up, etc., because of the ease of imaging.

There are large individual differences in FI values, even in the NOM. It has been reported that the FI of SCC in keratinised mucosa is significantly lower than that in non-keratinised mucosa [19]. Although there are still many items to be considered for the quantification of FI, we believe that much information can be obtained by standardising imaging techniques and quantifying and sharing data among institutions.

The NOM group used in the present study was significantly younger than the comparison group because of the cooperation of younger volunteers; however, no comparison of FI values or CV of NOM by age was made, which is also a limitation of this paper. The site of imaging, age, and individual differences in the FI values of healthy mucosa will be investigated in the future.

## 5. Conclusions

In this study, we photographed OMLs and NOM using IllumiScan^®^ and measured their FI values. Our data indicated that by analysing the FI and its CV obtained with the IllumiScan^®^, a differential diagnosis of SCC can be made among OMLs.

## Figures and Tables

**Figure 1 ijerph-19-10414-f001:**
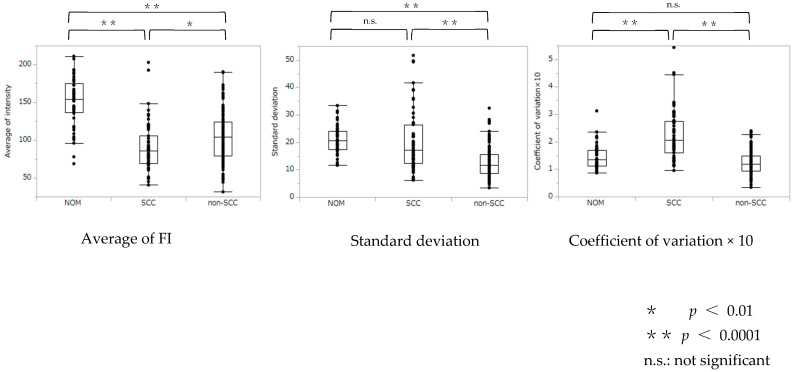
Analysis of variance comparing average fluorescence intensity, standard deviation, and coefficient of variation × 10 in normal oral mucosa (NOM), squamous cell carcinoma (SCC), and oral mucosal lesions except SCC (non-SCC).

**Figure 2 ijerph-19-10414-f002:**
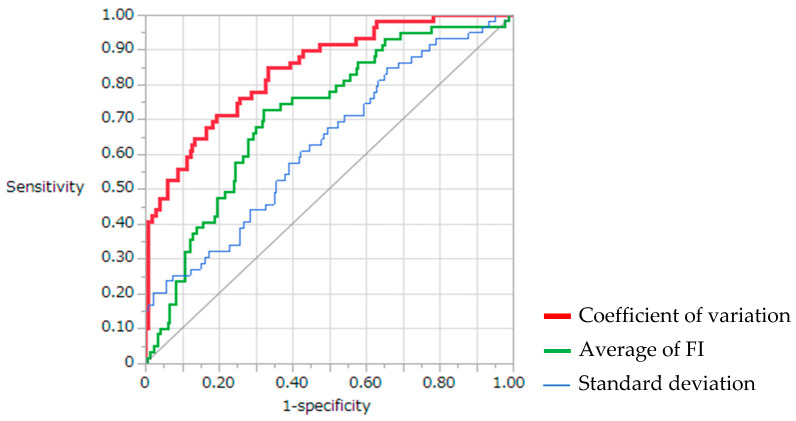
Objective evaluation based on receiver operating characteristic curve analysis. FI, fluorescence intensity.

**Table 1 ijerph-19-10414-t001:** Clinicopathologic characteristics of patients.

Characteristics	SCC	Non-SCC	NOM
Total cases	59	131	49
Male	35	50	30
Female	24	81	19
Median age (range), year	75 (39–98)	70 (20–98)	36 (25–94)
Median area, pixels	5.13 × 10^4^	2.09 × 10^4^	8.50 × 10^4^
Topographic location			
Buccal	4	53	11
Gum	29	42	6
Tongue	21	34	26
Floor of the mouth	5	0	6
Palate	0	2	0
Pathological diagnosis			
Squamous cell carcinoma	59		
Carcinoma in situ		16	
Epithelial dysplasia		22	
Hyperplasia		10	
Oral lichen Planus		64	
Inflammatory lesion		19	

SCC, squamous cell carcinoma; non-SCC, oral mucosal lesions except SCC; NOM, normal oral mucosa.

**Table 2 ijerph-19-10414-t002:** Univariate and multivariate analysis of the comparison between SCC and non-SCC.

	Univariate Analysis	Multivariate Analysis		
	*p*-Value	*p*-Value	Odds Ratio	95% CI
Sex, male/female	0.0033 *	0.39	1.43	0.63–3.25
Age	0.16	0.31	1.02	0.98–1.05
Average of intensity	0.013 *	0.57	0.99	0.95–1.03
Standard deviation	<0.0001 **	0.95	0.99	0.78–1.26
Coefficient of variation × 10	<0.0001 **	0.038 *	13	1.09–154.7

* *p* < 0.05; ** *p* < 0.01; SCC, squamous cell carcinoma; non-SCC, oral mucosal lesions except SCC.

## Data Availability

Not applicable.

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
