# Peer review of "Evaluation of Oral Mucosal Lesions Using the IllumiScan® Fluorescence Visualisation Device: Distinguishing Squamous Cell Carcinoma"

_ijerph, 2022, doi:10.3390/ijerph191610414_

Round 1
Reviewer 1 Report
The content of the manuscript regarding content and methodology is relevant and well presented. The technology studied is an important step for SCC diagnosis. My concern, thought, is that authors imply that the FV could replace the gold standard toll for oral cancer diagnosis (biopsy) when many limitations of the technique were described by the own manuscript.
I suggest authors to clarify that biopsy in line consequences described in line 42/43 ("seed 42 tumour cells into the surrounding tissues, leading to cancer recurrence and treatment failure" ) are not common, and to review the sentence "This may even replace the gold-standard biopsy as a diagnostic method." in line 287, in the Discussion section.
Author Response
Reply:
We thank the reviewer for pointing out the limitations of FV. We would like to clarify that we did not mean to imply that FV can replace biopsy. We have made the necessary changes to the manuscript accordingly in line 41-43 and 300-308.
Reviewer 2 Report
Interesting article, conclusions should be expanded and more current literature should be added ( up to 5 years back).
Author Response
Reply:
We thank the reviewer for pointing this out. We have added the relevant paper accordingly.
Reviewer 3 Report
The present study evaluated whether fluorescence intensity (FI) and its coefficient of variation (CV) can be used to diagnose squamous cell carcinoma (SCC) through IllumiScan®, an oral mucosa fluorescence visualisation (FV) device. The authors investigated 190 patients with oral mucosal lesions (OMLs; SCC, 59; non-SCC OMLs, 131) and 49 patients with normal oral mucosa (NOM), and then, compared the average FI, standard deviation (SD), and CV using IllumiScan®. Generally, the experiment design was reasonable. However, some problems have also been found in this manuscript. Some are detailed as follows for the authors' consideration:
“Introduction”- “Several devices have been developed and commercialised to evaluate tissue autofluorescence, such as the VELscope® (LED Dental Inc., Vancouver, Canada) and the IllumiScan® (Shoufu, Kyoto, Japan).” There was no description about VELscope®. The authors should supplement appropriately.
“The advantage of autofluorescence devices is that they enable the direct visualisation of tissue fluorescence; thus, there are many reports of visual recording and subjective judgment.” Please give some examples about the reports of visual recording and subjective judgment.
There were some similar studies that have evaluated the the IllumiScan fluorescence visualization device in detecting oral mucosal lesions (Kikuta S, et al, 2018; Morikawa T, et al, 2020). Please specify the innovation of this study.
“Materials and Methods”- Please add the inclusion and exclusion criteria of the clinical cases selected in this study.
The authors selected the cases that were found to have features compatible with the clinical suspicion of OPMD or SCC, not the confirmed cases. Does it affect the credibility of the study?
“The brightness of the irradiation light was maintained and did not vary between cases. The distance between the IllumiScan® lens and lesion was approximately 30 mm.” Whether the different brightness of the radiation light and the distance between the IllumiScan® lens and lesion will affect the fluorescence intensity (FI)?
“The ROIs of the NOM areas were set at the sublingual mucosa, buccal mucosa, the transition area from the molar attachment gingiva to the buccal mucosa and the mucosa on the floor of the oral cavity.” Please provide the reference.
“Results”- The results of cut-off values were missing.
The description of Table 2 was brief. The authors should add more detailed information.
“Discussion”- “The relationship between FI values and histopathological features has been examined, and the association between the mean width of keratin and hypofluorescent and hyperfluorescent carcinomas has been reported.” Please elaborate the “the relationship” and “the association” appropriately.
“This is also supported by the fact that the CV was independently significant in the multivariate analysis.” Please provide the reference.
The effects of sex and age on SCC and non-SCC groups were absent, and should be discussed.
The authors should improve the design of this experiment when realizing the potential limitations.
“References”-- Nearly half of the literatures cited were published five years ago. The authors should follow up the new progress in this field.
Author Response
“Introduction”- “Several devices have been developed and commercialised to evaluate tissue autofluorescence, such as the VELscope® (LED Dental Inc., Vancouver, Canada) and the IllumiScan® (Shoufu, Kyoto, Japan).” There was no description about VELscope®. The authors should supplement appropriately.
Reply:
We thank the reviewer for pointing this out. We have made the following addition to the manuscript accordingly:
“Although many devices have been developed and commercialized to evaluate tissue autofluorescence, the VELscope® (LED Dental Inc., Vancouver, Canada) was the first commercial auto-focus imaging device approved for oral use in 2008 [4,17]”.
“The advantage of autofluorescence devices is that they enable the direct visualisation of tissue fluorescence; thus, there are many reports of visual recording and subjective judgment.” Please give some examples about the reports of visual recording and subjective judgment.
Reply:
We thank the reviewer for pointing this out. We have made the following addition to the manuscript in response to this comment:
“For example, Ganga et al. visually classified lesions into two groups: a group containing lesions that showed FVL, appeared dark with pale green autofluorescence compared to surrounding unchanged tissue, and showed malignant or dysplastic changes and another group that showed fluorescence visualization retention (FVR) and no autofluorescent changes compared to surrounding unchanged tissue [20]. Meleti et al. subjectively classified lesions as normofluorescent, hypofluorescent, or hyperfluorescent [18]. Wang et al., after determining FVL, FVR, and fluorescence visualization increase (FVI), further classified them according to autofluorescence pattern into FVR, FVL+FVR, FVI, and FVI+FVL [17]”.
There were some similar studies that have evaluated the the IllumiScan fluorescence visualization device in detecting oral mucosal lesions (Kikuta S, et al, 2018; Morikawa T, et al, 2020). Please specify the innovation of this study.
Reply:
We thank the reviewer for pointing this out. Subjective judgments relying solely on visual perception introduce biases such as the observer's experience and judgment. In order to make this FV device versatile, quantitative analysis was necessary. Kikuta et al. and Morikawa et al. used luminance rate, and Kozakai et al. (39) used a border change rate. The novelty of our study is that we compared luminance and its variation by setting a healthy region to a region similar to that of the lesion.
“Materials and Methods”- Please add the inclusion and exclusion criteria of the clinical cases selected in this study.
Reply:
We thank the reviewer for pointing this out. We added the following in response to this comment:
“Inclusion criteria were patients who gave informed consent for imaging, were imaged prior to biopsy, and had a definitive diagnosis confirmed by biopsy. Patients with lesions of T3 or higher as per the International Union Against Cancer (UICC) eighth edition TMN malignancy classification and patients without adequate images due to aperture defects were excluded from this study to reduce the burden on patients [23]”.
The authors selected the cases that were found to have features compatible with the clinical suspicion of OPMD or SCC, not the confirmed cases. Does it affect the credibility of the study?
Reply:
We thank the reviewer for pointing this out. The aim of this study was to determine whether the investigational device could be used for OPMD and early invasive cancer. We believe that our study has a certain degree of credibility because all patients who visited our department with clinically suspected OPMD, who provided informed consent, and who were eligible (other than those with impaired opening or those who were not suitable for T3 or higher imaging) were included in the study.
“The brightness of the irradiation light was maintained and did not vary between cases. The distance between the IllumiScan® lens and lesion was approximately 30 mm.” Whether the different brightness of the radiation light and the distance between the IllumiScan® lens and lesion will affect the fluorescence intensity (FI)?
Reply:
We thank the reviewer for pointing this out. Several papers state that the distance between the lens and the lesion is 100 mm. Compared to these, the distance between the lens and the lesion in this study was small. Therefore, the intensity of irradiation light and excitation light may be larger. The FI may also be larger, although this is a relative value.
“The ROIs of the NOM areas were set at the sublingual mucosa, buccal mucosa, the transition area from the molar attachment gingiva to the buccal mucosa and the mucosa on the floor of the oral cavity.” Please provide the reference.
Reply:
We thank you for pointing this out. We have added the relevant reference accordingly.
“Results”- The results of cut-off values were missing.
Reply:
We thank the reviewer for pointing this out. We have added cut-off values accordingly.
The description of Table 2 was brief. The authors should add more detailed information.
Reply:
We thank the reviewer for pointing this out. We have added a more detailed description of Table 2 as follows: On the other hand, multivariate analysis revealed that only the CV (p=0.038, Odds ratio=13, 95% CI: 1.09–154.7) was independently significant (Table 2)
“Discussion”- “The relationship between FI values and histopathological features has been examined, and the association between the mean width of keratin and hypofluorescent and hyperfluorescent carcinomas has been reported.” Please elaborate the “the relationship” and “the association” appropriately.
Reply:
We thank the reviewer for pointing this out. We have added the following content accordingly:
“In a report examining histological characteristics, the epithelium of malignant lesions with a thick keratin layer was reported to be hyperfluorescent, while lesions with a thin keratin layer were dark and hypofluorescent [18]. The FI value of SCC was thought to capture the thickness of keratin, not just the attenuation of autofluorescence. Furthermore, considering that hyperplasia and increased keratinization are frequently observed in OPMD such as leukoplakia and in highly differentiated SCC, the FI value could also indicate the differentiation level of OSCC, since it has been reported that highly fluorescent lesions are in the early stages of malignant transformation, while hypofluorescent lesions are found at the undifferentiated cancer-like stage [34].”
“This is also supported by the fact that the CV was independently significant in the multivariate analysis.” Please provide the reference.
Reply:
We thank the reviewer for pointing this out. We have added the relevant relevance accordingly.
The effects of sex and age on SCC and non-SCC groups were absent, and should be discussed.
Reply:
We thank the reviewer for pointing this out. Supplemented.
This time, the coefficient of variation of luminance was independently significant. Therefore, regardless of the age and gender, I was able to suggest that it is a lesion. However, we also found a mean difference of 5 years between SCC and non-SCC because patients with inflammations such as lichen planus were younger than those with cancer. Assuming that the mucosa changes with age and that autofluorescence also changes, an age-matched comparison may be necessary.
Regarding the gender difference, we need to consider the difference in preference between men and women, which we consider to be a limitation of this study.
“References”-- Nearly half of the literatures cited were published five years ago. The authors should follow up the new progress in this field.
Reply:
We thank the reviewer for pointing this out. We have added relevant references accordingly.